# New Insights into the Role of Columellar Strut and Septal Extension Graft: A Comparative Review of Long-Term Outcomes

**DOI:** 10.3390/diagnostics15162051

**Published:** 2025-08-15

**Authors:** Riccardo Nocini, Nicola Magagnotto, Salvatore Chirumbolo, Massimo Albanese, Dario Bertossi

**Affiliations:** 1Unit of Otolaryngology, Head and Neck Department, School of Medicine, University of Verona, 37134 Verona, Italy; riccardo.nocini@aovr.veneto.it; 2Unit of Dental and Maxillofacial Surgery, Department of Surgery, Dentistry, Pediatrics and Gynecology, University of Verona, 37134 Verona, Italy; massimo.albanese@univr.it (M.A.); dario.bertossi@univr.it (D.B.); 3Department of Engineering for Innovation Medicine, University of Verona, 37134 Verona, Italy; salvatore.chirumbolo@univr.it

**Keywords:** columella strut, graft, nose, rhinoplasty, septal extension

## Abstract

**Background**: The columellar strut and septal extension graft are two common techniques in modern rhinoplasty to enhance nasal tip projection and rotation, though their effectiveness varies. While numerous studies examine each technique individually, there is limited research directly comparing the two methods, especially regarding long-term outcomes. This study aims to clarify the roles of these grafting techniques and possibly compare their long-term effectiveness in reconstructing caudal nasal support structures with available evidence in the literature. **Methods**: A comparative study was conducted on 87 patients with structural and anatomical nasal defects who underwent rhinoplasty using either a columellar strut or a septal extension graft. Outcome variables included tip projection, tip rotation, and nasal length. Patients were evaluated pre-operatively and at intervals of 1 month, 6 months, 1 year, and 4 years post-operatively. A critical discussion was also conducted to define the roles of the two nasal grafts and assess their long-term effectiveness. **Results**: Although over 4000 publications address the use of grafts in rhinoplasty, only a few compare the two grafting techniques directly. Our analysis revealed mixed perspectives on the columellar strut: some argue it is unnecessary, while others suggest it is not superior to septo-columellar suturing techniques. Conversely, several studies highlight the septal extension graft’s importance in reconstructing nasal support, with some evidence indicating it outperforms the columellar strut in maintaining long-term structure. **Conclusions**: Both grafts contribute to increased nasal projection; however, the septal extension graft provides more stable, long-term tip position support. The columellar strut appears non-essential for enhancing tip projection and should be used selectively, as it increases projection only when specifically designed for that purpose by the surgeon.

## 1. Introduction

Nasal tip support is a highly debated aspect of rhinoplasty, as reshaping the nasal area often involves cartilaginous grafts that play a vital role in achieving structural stability and optimal projection. About 1095 papers deal with “nasal tip support” and more than 4000 use the term “use of grafts in rhinoplasty” in PubMed, emphasizing the prominence of this topic in maxillofacial surgery. The positioning, control, and maintenance of the nasal tip are essential for successful rhinoplasty outcomes and are influenced by the surgical approach, technique, and post-surgical recovery [1,2,3,4,5,6,7,8,9,10,11,12,13,14,15,16,17,18,19,20,21,22,23].

As the nose functions to warm, filter, and hydrate inhaled air, maintaining proper columellar structure and tip support is important for both functional and aesthetic outcomes [23,24]. Two commonly used methods to stabilize and reshape the nasal tip are the columellar strut graft (CSG) and the septal extension graft (SEG). These grafts provide support and facilitate precise adjustments in nasal tip projection and rotation, contributing to the longevity and predictability of surgical results.

The nasal tip and base are supported by several anatomical elements (Figure 1), including the lower lateral cartilages (LLCs), the ligamentous attachment of the medial crura to the septal cartilage, the interdomal ligament bundle, the membranous septum, the anterior nasal spine, a skin envelope, and the lateral crural attachment (ACL) to the piriform aperture (PA) [25,26]. Preservation of these structures during rhinoplasty is vital, but certain surgical maneuvers may necessitate their reconstruction to restore nasal contour and function (Figure 2).

Over recent decades, grafting techniques like CSG and SEG have been developed to enhance the nasal framework, especially in cases requiring revision or secondary rhinoplasty. These grafts serve key roles in managing tip rotation and projection and can be tailored to the patient’s anatomical needs.

Despite their widespread use, long-term comparative data evaluating CSG and SEG remain limited. While many studies examine each technique in isolation, few directly compare their performance in maintaining nasal tip projection and rotation over extended follow-up. Therefore, the present study aims to provide new insights by assessing and comparing the long-term effectiveness of these two grafting methods in patients undergoing rhinoplasty (Table 1).

## 2. Materials and Methods

### 2.1. Patients

Rhinoplasty interventions were performed at the UOC of Maxillofacial Surgery of the University Hospital of Verona. A total of 100 patients were considered for this study; however, 13 patients did not complete the study and/or did not meet the inclusion criteria, and were therefore excluded. The major inclusion criteria were patients presenting with clinically and/or radiographically confirmed structural or anatomical nasal defects—such as asymmetrical, under-projected, or under-rotated nasal tips—that warranted surgical correction. These conditions were assessed by a senior maxillofacial surgeon (DB) through pre-operative clinical examination, supported by photographic analysis and, where necessary, radiologic imaging. Eligibility for inclusion in this study was determined based on the patient’s need for graft-based nasal tip support reconstruction as part of a planned rhinoplasty.

The exclusion criteria included severe nasal septal deviation or significant nasal airway obstruction, in addition to the presence of systemic conditions such as metabolic, cardiologic, dermatologic, or oncologic diseases. These conditions were excluded due to their potential impact on wound healing, post-operative recovery, and surgical risk, which could compromise the consistency and reliability of outcome assessments.

All patients underwent the surgical intervention by the same surgeon (DB) between January 2016 and January 2020, including nasal basal osteotomies, tip projection, tip rotation, and intranasal graft placement, where the primary variable was the use of a *columellar strut* (n = 46) or *septal extension graft* (n = 41).

All pre- and post-operative photographs were taken by the same trained operator using a Canon EOS 5D Mark IV camera with a Canon EF 100 mm f/2.8 Macro USM lens (Canon Inc. Ōta, Tokyo, Japan). To ensure reproducibility, images were captured under consistent lighting conditions with the patient’s head positioned naturally and aligned to a standardized facial grid referencing the true vertical line.

The selection of graft type (CSG vs. SEG) was based on the operating surgeon’s preference, guided by patient-specific clinical indications such as anatomical structure, cartilage strength, and desired tip modification. This non-randomized assignment introduces a potential for selection bias, which is acknowledged as a study limitation.

From a statistical point of view, 87 patients are needed to meet the statistical criteria to prevent size effects in sampling, as 72 is the minimum to achieve a 5% error (within CI_95%_) and a proportion population of 50%. To reach a percentage of success in the treated population at least close to 95%, the inclusion of 87 patients allows the surgeon to reach a margin of error <5% (4.58%).

Patients underwent anamnaestic and pre-clinical surveys, according to our internal procedures and current guidelines, before entering this study.

The study is a secondary analysis based on publicly available NHANES data. NHANES data collection procedures were approved by the National Center for Health Statistics Research Ethics Review Board, and all participants provided informed consent in accordance with their protocols.

### 2.2. Analysis

The primary outcome measures were nasal tip projection and tip rotation. The secondary outcomes included nasal length and changes in nasolabial angle over time. All measurements were conducted by an independent rater who was blinded to the graft type to reduce assessment bias. Photographic evaluations were performed at five intervals: pre-operatively and at 1 month, 6 months, 1 year, and 2 years post-operatively using a Canon EOS 5D Mark IV with a Canon EF 100 mm f/2.8 Macro USM lens. To ensure consistency, all images were captured by the same operator under standardized lighting and patient head positioning, aligned to a facial grid referencing the true vertical line [27,28].

Photographic evaluations were conducted at five intervals: pre-operatively and then at 1 month, 6 months, 1 year, and 2 years post-operatively, using a Canon EOS 5D Mark IV with a Canon EF 100 mm f/2.8 Macro USM lens (Canon Inc., Ōta, Tokyo, Japan). To ensure consistency, all photos were taken with the patient’s head in a natural position, and measurements were aligned to a facial grid with reference to the true vertical line [27,28]. Outcome variables included tip projection (measured linearly from the subnasale to the tip and assessed using Goode’s ratio) and tip rotation (evaluated by the nasolabial angle). Nasal projection was measured from the pyriform aperture to the pronasale, while nasal length was taken from the nasal root to the pronasale. The nasolabial angle was defined using three reference points: the tip, subnasale, and cupid’s bow. The results were organized by technique and time of measurement.

### 2.3. Surgical Techniques: General Procedure

All rhinoplasty procedures were performed using standardized techniques by the same surgeon (DB), employing either an open or closed approach based on individual case requirements. Initial steps involved careful dissection and mobilization of nasal structures, including release of the lower lateral cartilages and elevation of mucoperichondrial flaps. Dorsal reduction and septal cartilage harvesting (Figure 3 A-K) were performed as needed. Graft placement was the primary variable distinguishing the two patient groups.

The key surgical maneuvers and approaches to rhinoplasty were consistent across both patient groups. Both columellar strut graft (CSG) and septal extension graft (SEG) (Figure 4) procedures were performed using either an open or closed approach, involving careful separation of the nasal structures. All nasal components were first released, with additional maneuvers completed before graft positioning. Where possible, septal cartilage was harvested as the primary graft material, facilitated by dorsal reduction and elevation of the mucoperichondrial flaps.

For optimal aesthetic outcomes, columellar strut grafts were secured in a pocket positioned towards the anterior nasal spine and attached to the medial crura of the lower lateral cartilages. The CSG was anchored in an anterior pocket at the anterior nasal spine, secured to the medial crura using polydioxanone sutures and 4-0 polyglactin 910 sutures (for rapid absorption) through the vestibular skin on both the medial crura and the columellar strut. The SEG was secured to the septal L-strut cartilage and could be affixed in three areas: a suture connecting the septal extension graft to the caudal septum; a suture overlapping the cephalic border of the septal extension graft with the dorsal border of the septum (near the anterior septal angle); and a horizontal mattress suture securing the septal extension graft to the caudal lateral septum. The SEG overlaps one side of the caudal/lateral septum. Dome definition in both groups was achieved using intra-domal and inter-domal suturing of the middle crura of the lower lateral cartilages.

Both open and closed rhinoplasty approaches were utilized based on anatomical complexity and patient-specific requirements, as assessed by the senior surgeon. The choice of surgical approach did not determine graft selection, which was independently based on structural needs and clinical judgment. Open approach was more commonly used in anatomically complex cases requiring detailed graft placement.

In all cases where a septal extension graft (SEG) was used, the graft was consistently fashioned in an L-shape and affixed using a standardized set of three fixation points: the caudal septum, the anterior septal angle, and the caudal lateral septum. This approach was applied uniformly across patients to ensure procedural consistency and allow for reliable comparison of long-term outcomes.

### 2.4. Columellar Strut Graft (CSG) Technique

In the CSG group, a columellar strut was positioned in a precisely prepared pocket between the medial crura and anchored at the anterior nasal spine. Fixation was achieved using polydioxanone and absorbable polyglactin 910 sutures, ensuring stability through both the vestibular skin and medial crura. The graft was typically fashioned from septal or conchal cartilage, depending on intraoperative availability. Dome definition was refined using intra-domal and inter-domal sutures (Figure 5).

Figure 3 presents the sequential paths from left to right and from top to bottom of the surgical process of our study. CSGs and SEGs are typically autologous cartilaginous grafts, most commonly harvested from the septum, particularly when septal correction is required. Conchal cartilage is also a common source and, as demonstrated by Bucher et al., may provide better outcomes for upward rotation of the nasal tip [29]. For cases requiring greater nasal tip projection or correction of severe alignment abnormalities, grafts may be taken from resected autologous nasal bone or septal cartilage during nasal hump correction. The nasal bone, with its cortical, straight, and rigid structure, is especially effective in providing reliable tip support over time. In instances where neither septal nor conchal cartilage suffices for dorsal or tip support—often seen in cases of post-traumatic or congenital skeletal deficiencies—rib grafts are utilized to supply the necessary structure.

The columellar strut graft (CSG) is widely used to enhance nasal tip projection and improve the nasolabial angle in rhinoplasty. However, its effectiveness in maintaining long-term tip projection and rotation remains a topic of debate in the recent literature. Rohrich et al. specifically examined the CSG’s impact on tip projection, finding any effect to be minimal [30,31]. Additionally, Sadeghi et al., in a blinded randomized study in 2009, concluded that CSG is not essential for sustaining tip projection or rotation in rhinoplasty. A key limitation of the CSG is its lack of control over tip rotation, especially in patients with weak alar cartilage [32].

The columellar graft is an essential tool for achieving nasal tip stability, particularly during ligament dissection or in patients with weak, short, or asymmetrical medial crura. Situations warranting a CSG to support the tip include cases with an endonasal approach, limited graft material, adequate or excessive tip projection, or when only minimal tip position adjustment is needed [32,33,34,35,36,37,38,39,40]. Typically, the choice of CSG length and placement depends on the degree of nasal projection and the strength of the lower lateral cartilages; for instance, weak alar cartilages with adequate tip projection generally require a shorter CSG.

Managing columellar width is crucial, as excessive enlargement may yield an undesired aesthetic outcome, even when the primary aim is tip support. Notably, the lower lateral cartilage significantly influences the shape and character of the nasal tip and the lower third of the nose. Therefore, the CSG, given its integral connection to the medial and lateral crural cartilages (the defining points of the tip), plays a vital role in shaping nasal aesthetics when applied thoughtfully.

### 2.5. Septal Extension Graft (SEG) Technique

For the SEG group, the graft was sutured directly to the caudal septum and extended anteriorly to support the tip. Multiple anchoring points—caudal septum, anterior septal angle, and lateral caudal septum—were used to ensure structural rigidity. SEG positioning allowed for adjustment of both the tip projection and rotation. Cartilage was primarily harvested from the septum or, when inadequate, from costal cartilage.

The septal extension graft (SEG), introduced by Byrd in 1997, was designed to provide better control over nasal tip shape, projection, and rotation. Unlike the columellar strut graft (CSG), which may be less effective in cases of drooping tips or weak nasal valves, the SEG offers more robust structural support, making it the preferred choice when greater tip stability is needed.

There are various forms of SEGs, differing in shape and fixation methods, but all share common attachment points in the nasal lobule area. These points of fixation are chosen based on factors such as skin thickness, nasal size, and septal stability. For optimal effectiveness, the SEG should extend beyond the anterior septal angle into the interdomal space. This positioning allows the graft to sit in the columellar corner area, with the main fixation point at the divergence of the medial crura.

The SEG can be fashioned from nasal septum or costal cartilage, although a deviated septum may limit its use due to potential cartilage deformation. When correctly anchored with multiple sutures across nasal structures, the SEG offers greater support than a CSG (Figure 6). However, if not adequately secured, it may lead to unsatisfactory outcomes. While SEG is highly effective for suitable patients, it is generally not recommended for noses with heavy lower lateral cartilages or with normal to excessive tip projection.

### 2.6. Technical Comparison Considerations

While both grafts aim to enhance nasal tip projection and rotation, they differ in their biomechanical behavior and indications. CSGs are simpler to place and provide passive support, often suitable for minor tip repositioning. SEGs offer greater control over tip dynamics but require more precise alignment and multiple fixation points. Choice of graft was standardized across patient groups to enable a direct comparison of long-term structural outcomes.

### 2.7. Statistics

Statistics for repeated measures like the ones reported in Table 2, the mixed-effects ANOVA (or repeated measures ANOVA), were applied. The factors considered were a) group (CSG vs. SEG) → between-subject factor and b) time (pre-op, T1, T2, and T3) → within-subject factor. The interactions were group × time interactions to see if the evolution over time differs between groups. The effect size (Cohen’s d) for key pairwise comparisons (e.g., pre-op vs. T3 within each group) was evaluated. A trend analysis, which can identify if there is a significant linear or quadratic trend over time for each variable, was conducted. The parameters considered were a) nasal length, where CSG increased at 1 month, then trended back toward baseline by 24 months, and SEG underwent a similar increase at 1 month but was slightly higher at 24 months than at baseline, leading to the interpretation of a temporary increase post-op with partial regression to baseline, where SEG may retain a small increase; b) nasolabial angle, where CSG showed a notable increase post-op, with some reduction over time but still above baseline at 24 months, and SEG reported an increase post-op but remained more stable and higher than pre-op even at 24 months, suggesting the interpretation that surgery’s effect on angle is more stable in SEG, with less regression. To obtain a much more robust statistic, we also applied a linear mixed-effects model (LMM) to account for individual variability and checked the sphericity in repeated measures ANOVA (if violated, applied corrections like Greenhouse–Geisser). LMM is used for repeated measures data, where each patient contributes multiple observations over time. It accounts for within-subject correlation.

$$\mathrm{Nasolabial}\;{\mathrm{angle}}_{i,j}=\beta_{0}+\beta_{1}\cdot {\mathrm{Group}}_{i}+\beta_{2}\cdot ({\mathrm{Group}}_{i}\cdot {\mathrm{Time}}_{i})+u_{i}+\varepsilon_{i,j}$$where *i* = patient; *j* = time point; *β*_0_ = intercept, CSG, pre-op; *β*_1_ = effect of SEG vs. CSG; *β*_2_ = effect of time (T1, T2, and T3); *β*_3_ = interaction effects (group × time); *u_i_* = random effect for patient *i*; and *ε_i_*_,*j*_ = residual error. The coefficients *β* are estimated using Restricted Maximum Likelihood (REML). These are differences in group means adjusted for other variables. For example, *β*_1_ = 2.768 means the SEG group starts ~2.77° higher than CSG.

The model includes a random intercept per patient to capture individual baseline variability:$$u_{i}∼N\;\left(0,\;\sigma_{i}^{2}\right)$$

This prevents pseudoreplication—falsely treating repeated measures as independent.

*p* values, F-statistics, and confidence intervals were reported, and effect sizes were included for clinical relevance. The rotation of the nasal tip holds significant clinical relevance because it directly influences both the aesthetic profile and functional airway dynamics of the nose. From an aesthetic perspective, proper tip rotation is crucial in defining nasal harmony and balance, particularly in relation to the nasolabial angle and facial proportions. Excessive upward or downward rotation can distort facial symmetry and produce an unnatural appearance. Functionally, inappropriate tip rotation may affect the external nasal valve, potentially leading to compromised airflow or nasal obstruction. Thus, achieving and maintaining optimal tip rotation is essential for both patient satisfaction and physiological function.

Although both CSG and SEG are widely used and well-documented, there is limited long-term comparative data on their relative effectiveness. Our literature review identified only three studies directly comparing the two grafts in terms of long-term outcomes. Furthermore, there is a notable absence of large-scale systematic reviews or meta-analyses specifically addressing their comparative performance. This highlights a significant gap in the evidence base, underscoring the need for focused investigations such as the present study, which aims to provide clarity on the long-term effectiveness of these techniques.

Limitations: due to the limited number of samples, statistical significance was not optimally achieved, though the results show the correct procedure and the good outcome of our surgical procedures. The results are reported in Figure 1, Figure 2, Figure 3, Figure 4, Figure 5 and Figure 6.

## 3. Results

Figure 1, Figure 2, Figure 3, Figure 4, Figure 5 and Figure 6 show our surgical procedure. To the best of our knowledge, and based on our review of PubMed, Scopus, WoS, and other databases, we identified 3376 publications on nasal tip grafts. Of these, 1096 addressed nasal grafts, 176 focused on columellar strut grafts (CSG), 183 focused on septal extension grafts (SEG), 35 discussed both CSG and SEG, and only 3 directly compared CSG and SEG.

From this analysis, it is clear that while there is a significant amount of studies on various grafting techniques in rhinoplasty, studies directly comparing CSG and SEG remain limited. Only three studies specifically examine the comparative effects of these two grafts in correcting structural nasal defects, such as under-projected tips. The 35 articles that discuss both CSG and SEG are generally focused on aspects like surgical techniques, complications, and individual outcomes rather than comparative effectiveness. In several studies, SEG has demonstrated a role in reconstructing nasal support structures, and in the few available comparative studies, it has shown some superiority over CSG.

### Long-Term Follow-Up

We studied 87 patients (62% female and 38% male) with an average age of 38 years, of whom 46 received a CSG and 41 an SEG. All patients were followed for 24 ± 2 months. Using photogrammetric analysis at multiple post-operative intervals, we compared outcomes between the CSG and SEG groups. Long-term assessments measured nasal length relative to the true vertical line (Figure 7) and the nasolabial angle (Figure 8).

Our analysis showed statistically significant changes in tip position and rotation immediately post-operatively (1 month) and in the later post-operative period (12–24 months). Specifically, we observed a decrease in both tip position and rotation in the first month post-surgery for both CSG and SEG cases. However, after this period, differences emerged between the groups: the CSG group showed a more pronounced decrease in tip stability over the first year, while the SEG group maintained greater tip stability. In the second post-operative year, minor further changes in tip position were observed in the CSG group (Table 2 and Table 3).

Key results from the linear mixed-effects model analyzing nasolabial angle changes over time showed the following. (a) Intercept (CSG, Pre-op): 93.53°. This is the baseline nasolabial angle for the CSG group at pre-op. (b) SEG vs. CSG (Main Effect): +2.77°, *p* = 0.049, which is statistically significant. SEG starts with a higher angle than CSG. Regarding time effects (within CSG groups), (a) T1 (1 month): +11.74°, *p* < 0.001; (b) T2 (12 months): +6.52°, *p* < 0.001; (c) T3 (24 months): +5.93°, *p* < 0.001. All show statistically significant increases from pre-op. Moreover, regarding group × time interactions (SEG relative to CSG): a) T1: −4.28°, *p* = 0.030, so SEG has a significantly smaller increase at 1 month than CSG. T2 and T3 were not statistically significant. The SEG group starts higher but increases less sharply at 1 month than CSG. At 24 months, differences between SEG and CSG are not statistically significant, suggesting convergence or stability. We achieved *p* < 0.05 for two key contrasts: (a) SEG vs. CSG overall; (b) SEG vs. CSG at 1 month.

## 4. Discussion

Despite a systematic review and meta-analysis of primary and secondary studies not being the major target of our research study, our results were put in comparison with previous and valuable approaches, to reach a final estimation of the impact of our study. The purpose was to compare the effectiveness of columellar strut (CSG) and septal extension graft (SEG) in maintaining medium- and long-term nasal tip projection and rotation. This analysis focused on patients undergoing primary, secondary, or revision rhinoplasty using either open or closed techniques to correct structural and anatomical nasal defects. Searches were conducted in PubMed and major databases, including the Cochrane Database of Systematic Reviews and ResearchGate, from January to June 2021, covering studies published over the past 30 years.

The inclusion criteria were participant type (any age and both sexes), study type (clinical trials, meta-analyses, and systematic reviews), intervention type (primary, secondary, or revision rhinoplasty using CSG or SEG, or osteotomies using piezosurgery [41,42,43]), and language (English or Italian). No studies were excluded due to limited publication data on this topic.

The columellar strut graft (CSG) and septal extension graft (SEG) are two widely used methods for controlling nasal tip projection and rotation in modern rhinoplasty. Both grafts offer the potential to increase tip projection and rotation, though the choice between CSG and SEG often depends on surgeon preference. To date, few clinical trials have directly compared the two methods. While fixed columellar struts are less commonly used in aesthetic rhinoplasty, floating columellar struts remain popular. Columellar struts are particularly effective for unifying and stabilizing the nasal tip in cases with weak medial or middle crura, asymmetry in the lower lateral cartilages, or short medial crura. However, they do have limitations, especially in controlling tip rotation, which is their primary drawback.

In noses with strong lower lateral cartilages or already sufficient tip projection, using a septal extension graft (SEG) may result in excessive stiffness, over-projection, or unnatural tip rotation. SEG offers powerful tip support but lacks the adaptability needed in these cases. This is supported by Byrd et al. (1997) [34] and Rohrich et al. (2012) [35], who recommend SEG mainly for under-projected or weak tip structures.

Byrd et al. introduced the SEG as a more reliable option for achieving tip projection, shape, and rotation control, proposing it as a means to redefine the structural relationship between the nasal tip and dorsum [34]. The role of the columellar strut in aesthetic nasal surgery continues to be debated, with limited studies specifically addressing its indications. This has fueled ongoing discussions regarding its effectiveness in maintaining tip projection and support.

The L-shaped configuration of the septal extension graft plays a crucial role in influencing surgical outcomes. By incorporating both vertical and horizontal limbs, the L-shape provides multidirectional support—enhancing nasal tip projection while also reinforcing dorsal stability. This configuration increases the rigidity of the nasal framework, offering more resistance against long-term forces such as scar contracture or tissue recoil. As a result, the L-shaped SEG contributes to more predictable and sustained tip positions, especially in cases requiring strong structural reinforcement. These biomechanical advantages make it particularly suitable for challenging cases where maximal support and durability are needed.

Rohrich, a key author in defining the role of CSG, developed a classification and proposed an algorithm for its application [35]. Analyzing a series of 100 consecutive rhinoplasty cases, he found that while CSGs do not necessarily increase tip projection, they are valuable for unifying the nasal tip and controlling its final position [36].

Sirinoğlu compared the effects of columellar strut grafts and septo-columellar sutures over a one-year follow-up, finding no statistically significant difference between the suture + graft group and the suture-only group [37]. Thus, CSGs may not consistently maintain desired tip rotations and projections, but they are useful for achieving significant projection when intended. Sadeghi et al. also investigated CSGs and found no specific role in sustaining tip projection and rotation after one year when used randomly in unselected cases [32]. Bucher et al. reported changes in nasolabial angles and nasal tip projections pre- and post-operatively following exclusive CSG implantation, observing that CSGs improved tip rotation and projection only when specifically intended by the surgeon [29].

Direct comparisons between CSG and SEG are rare, with only a few studies available. Akkus et al. conducted a comparative study on early- and long-term outcomes of nasal projection-to-length ratios and nasolabial angles in 36 patients, observing no significant differences between groups but noting that values decreased less in the SEG group [38].

Atighechi et al. performed an experimental study aimed at identifying the most effective method for correcting tip ptosis. They compared CSG and SEG techniques for addressing acute nasolabial angle (NLA) and columellar retraction. Although both methods provided satisfactory NLA and columellar correction, SEG proved more stable for patients with tip ptosis in long-term follow-up [39].

Perkins quantified the impact of tip rotation on upper lip length in a retrospective study of rhinoplasty patients with increased tip rotation, finding that the SEG group had greater stability over time and achieved a larger NLA and ULL compared to the CSG group [40]. A system to analyze and non-surgically treat the nose support has also been described with the aim of providing Hyaluronic Acid Injections [28,44].

Finally, while the tongue-in-groove (TIG) technique is a well-established method for nasal tip positioning and stabilization, its effectiveness may vary depending on patient-specific anatomical factors. In our study, we applied septal extension grafts (SEGs) in all cases—even those where TIG was performed—to ensure methodological consistency for the comparison between SEG and columellar strut graft (CSG) outcomes. This decision was informed by the need to standardize structural support across groups, particularly given the inclusion of patients with challenging tip anatomy. Furthermore, the findings of Sazgar et al. (2022) [45] demonstrate that combining SEG with TIG enhances the maintenance of tip rotation over time compared to TIG alone. This supports our rationale for using SEG routinely in such cases, where robust and lasting structural integrity is paramount [45].

## 5. Limitations of This Study

This study has some limitations that may impact the generalizability and robustness of its findings. Firstly, the relatively small sample size (87 patients) may limit the statistical power of the results, particularly for subgroup analyses comparing columellar strut graft (CSG) and septal extension graft (SEG) techniques. Although the sample met the minimum threshold for significance, a larger cohort could provide more precise estimates of long-term outcomes and improve the reliability of comparative analyses. Secondly, this study did not account for variations in skin thickness, nasal cartilage strength, or individual healing responses, which may influence graft performance and nasal tip stability over time. Another limitation is the reliance on a single surgeon performing all procedures, which could introduce a procedural bias, limiting the ability to generalize findings to other surgical approaches or skill levels. Additionally, although follow-up was conducted over a two-year period, longer-term studies would be beneficial to observe the stability of nasal tip projection and rotation in both graft types over extended periods. Finally, this study lacked a randomized design, and therefore, the potential for selection bias cannot be entirely ruled out. Future studies should aim to include randomized, multi-center trials with a larger, more diverse patient population to validate these findings.

## 6. Conclusions

The results of this study suggest that future research should focus on optimizing graft selection based on individual anatomical and structural characteristics, potentially leading to more personalized approaches in rhinoplasty that could enhance long-term nasal tip stability and patient satisfaction.

In rhinoplasty, support and remodeling of the toe has always been a compelling challenge due to the numerous structures involved in its appearance. Since 1997, when SEG was introduced, tip grafting has been discussed in the literature and the role of CSG has been revised. Both grafts showed a role in increasing nasal projection; however, septal extension graft proved to be a more permanent method of stabilizing the tip position in long-term follow-up, but at the expense of a greater sensation of nasal stiffness by the patient and a longer duration of post-operative edema. CSG showed a non-essential role in enhance tip projection, and it can increase it only if specifically intended by the surgeon; therefore, it should be used only in selected cases. While CSG is indicated in noses with weak medial or middle crura, asymmetrical lower lateral cartilages, and/or a short medial crural when tip re-positioning is sought, SEG is proposed as a method to redefine the nasal architecture between the nasal tip and dorsum, particularly in patients at high risk of loss of projection.

## Figures and Tables

**Figure 1 diagnostics-15-02051-f001:**
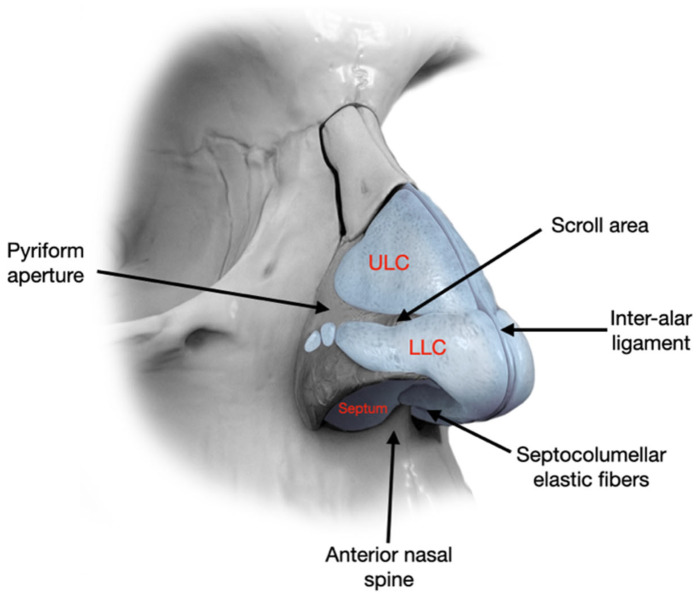
Fibrous and cartilaginous structures involved in nasal tip support.

**Figure 2 diagnostics-15-02051-f002:**
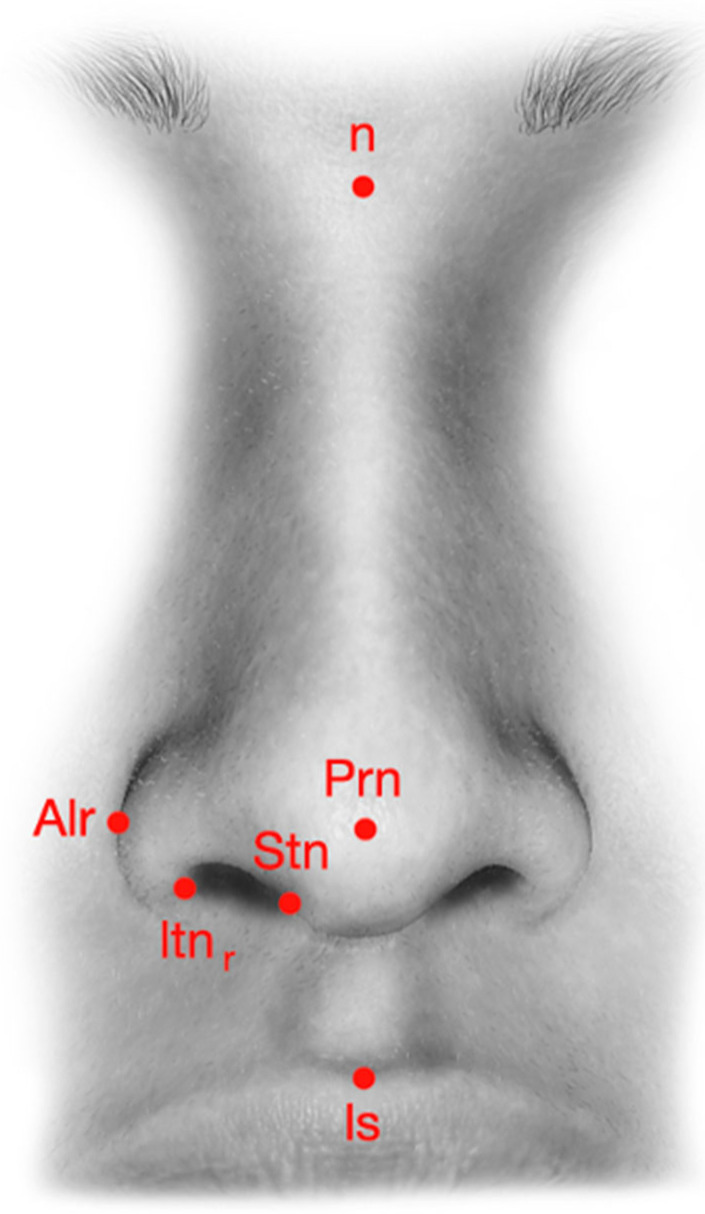
Nasal landmarks used in pre-operative analysis. Legends: n = nasion; Prn = pronasal; Stn = superior terminal of the nostril; Itn = inferior terminal of the nostril; Alr = alar; Ls = labiale superior.

**Figure 3 diagnostics-15-02051-f003:**
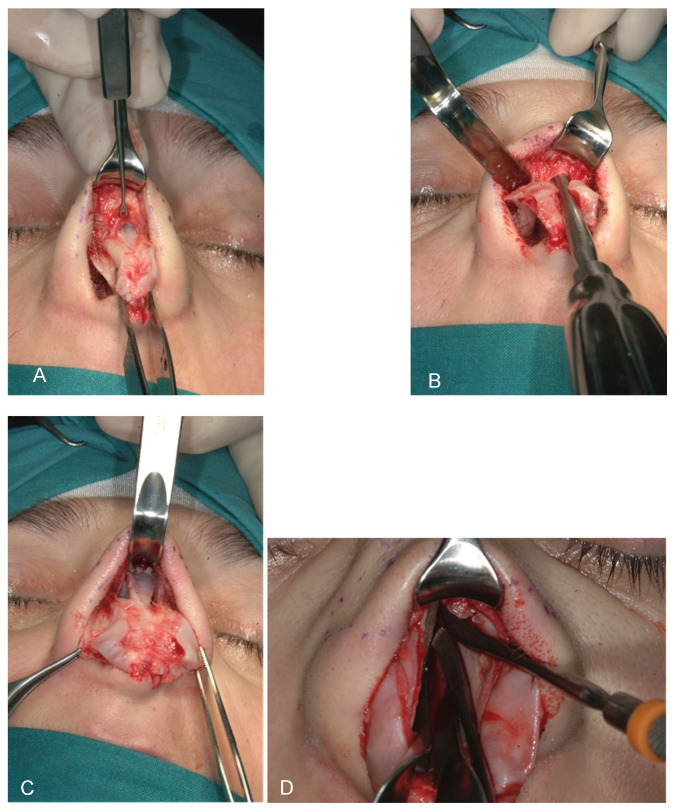
(**A**–**K**) from top and left to right: Photographic representation of septal cartilage removal (Prof. Dario Bertossi, Maxillo-Facial Surgery Verona, Italy).

**Figure 4 diagnostics-15-02051-f004:**
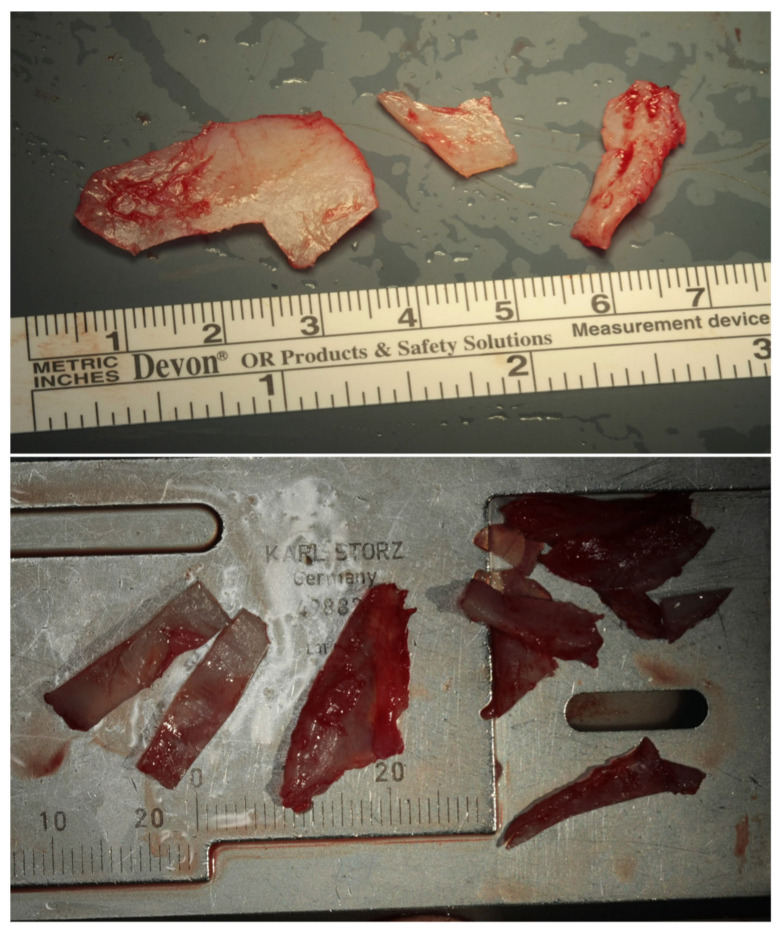
Intraoperative photos of a CSG and a SEG (Prof. Dario Bertossi, Maxillo-Facial Surgery Verona, Italy).

**Figure 5 diagnostics-15-02051-f005:**
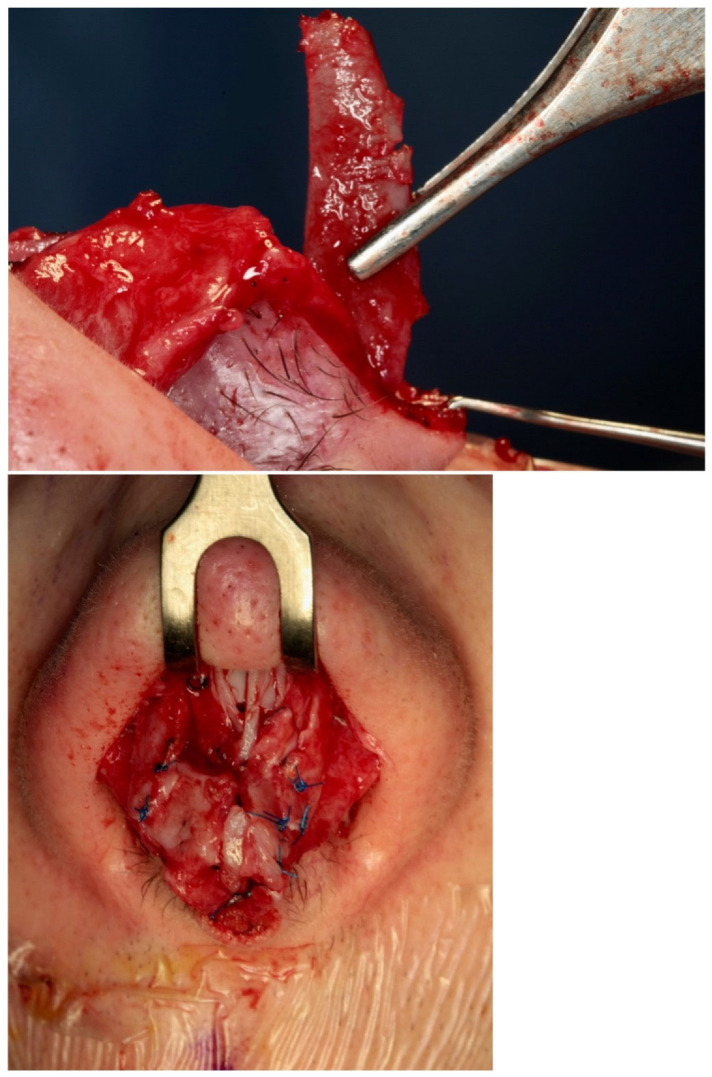
Intraoperative insertion and fixation of a columellar strut. Top: lateral view; bottom: frontal view (Prof. Dario Bertossi, Maxillo-Facial Surgery Verona, Italy).

**Figure 6 diagnostics-15-02051-f006:**
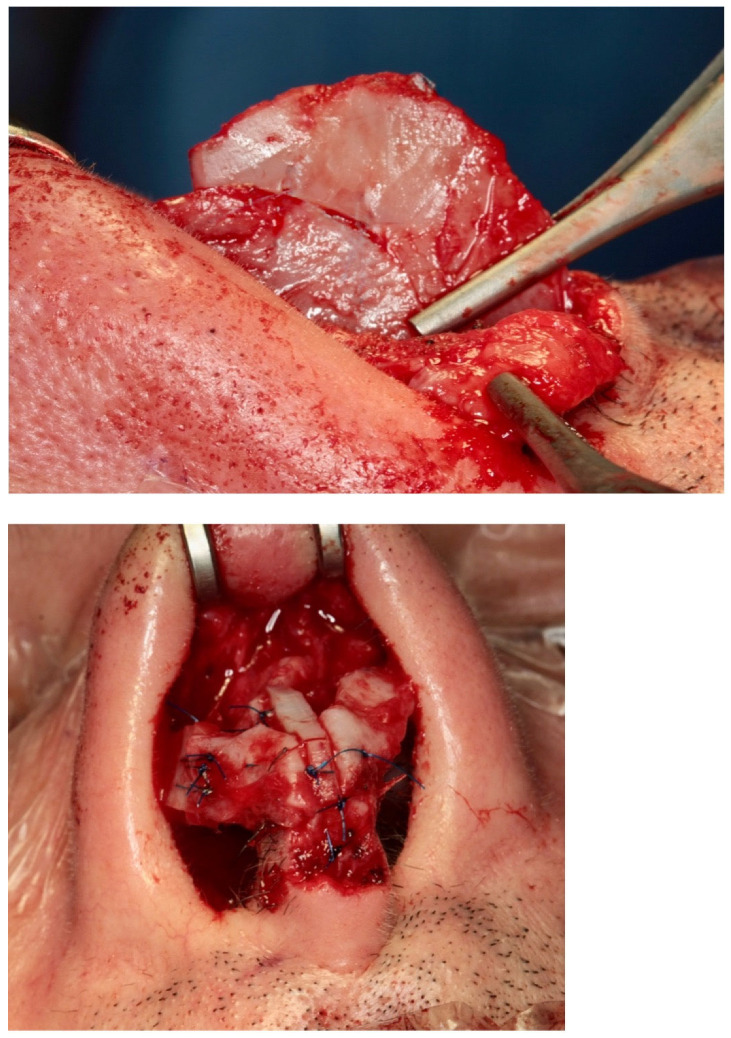
Intraoperative insertion and fixation of a septal extension graft. (Prof. Dario Bertossi, Maxillo-Facial Surgery Verona, Italy).

**Figure 7 diagnostics-15-02051-f007:**
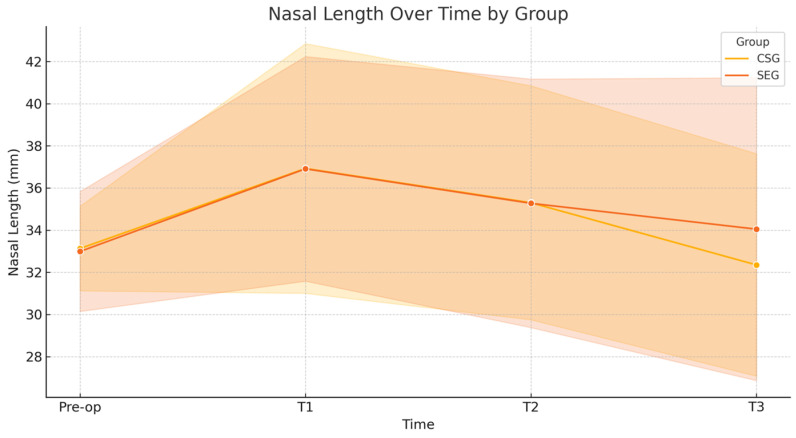
Nasal length over time. A linear mixed-effects model revealed no significant interaction between group and time for nasal length (*p* = 0.338). Both groups exhibited a transient increase in nasal length at 1-month post-operation, returning to near baseline values at 24 months. Effect sizes from pre-op to 24 months were small in both the CSG (d = 0.15) and SEG (d = −0.13) groups, indicating minimal long-term changes.

**Figure 8 diagnostics-15-02051-f008:**
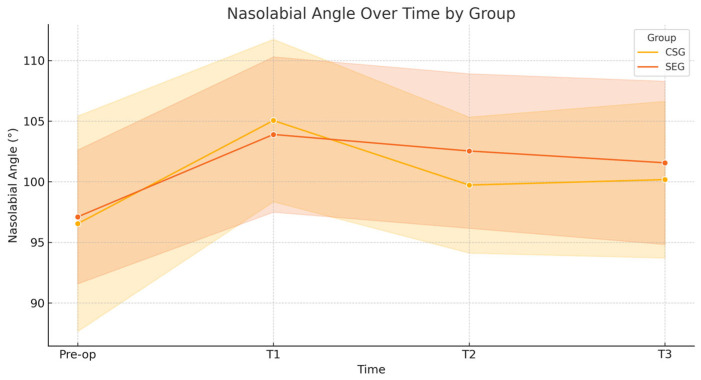
Nasolabial angle over time. For the nasolabial angle, the linear mixed-effects model indicated no significant interaction between group and time (*p* = 0.732). The CSG group showed a moderate decrease in angle from pre-op to 24 months (d = −0.35), while the SEG group demonstrated a larger moderate decrease (d = −0.64). These changes suggest a more stable long-term alteration in the nasolabial angle within the SEG group.

**Table 1 diagnostics-15-02051-t001:** Nasal landmarks.

Landmark	Definition
*Midline landmarks*	
Nasion (*n*)	Innermost point between the forehead and nose
Pronasal (*prn*)	Most protruded point of the nasal apex
Columella (*c*’)	Midpoint between the columella crests, level with the top of the corresponding nostril
Subnasal (*sn*)	Midpoint at the union of the lower border of the nasal septum and the upper lip
Labiale superius (*ls*)	Midpoint of the upper vermilion line
*Paired landmarks*	
Alar (*al*)	Most lateral point on each alar contour
Nasal alar crest (*ac*)	Most lateral point in the curved base of each alar
Inferior terminal of the nostril (*itn*)	Inferior point of the nostril axis
Superior terminal of the nostril (*stn*)	Superior point of the nostril axis
Alar (*al*)	Most lateral point on each alar contour

**Table 2 diagnostics-15-02051-t002:** Nasal length long-term follow-up analysis.

Nasal Length (mm)	Pre-Op	T1 (1 Month)	T2 (12 Months)	T3 (24 Months)
CSG	33.5 ± 3.8	36.2 ± 5.7	35.8 ± 5.5	33.9 ± 5.8
SEG	32.6 ± 3.3	35.4 ± 5.1	34.9 ± 5.0	34.5 ± 5.8

**Table 3 diagnostics-15-02051-t003:** Nasolabial angle long-term follow-up analysis.

Nasolabial Angle (°)	Pre-Op	T1 (1 Month)	T2 (12 Months)	T3 (24 Months)
CSG	94 ± 7.9	105 ± 6.7	101 ± 6.0	99 ± 6.1
SEG	96 ± 7.1	104 ± 6.9	103 ± 7.1	103 ± 7.2

## Data Availability

The original contributions presented in the study are included in the article, further inquiries can be directed to the corresponding author.

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
