# Peer review of "New Insights into the Role of Columellar Strut and Septal Extension Graft: A Comparative Review of Long-Term Outcomes"

_diagnostics, 2025, doi:10.3390/diagnostics15162051_

Round 1
Reviewer 1 Report (New Reviewer)
Comments and Suggestions for Authors
Authors presented a very interesting article about a controversial issue SEG versus strut. However , I think the below points can improve it:
1- Would you please explain "it is generally not recommended for noses with heavy lower lateral cartilages or with normal to excessive tip projection. " and if possible provide references for it
2- Authors use L shape SEG , which may have some effect on final results , which needs to be explained in discussion.
3- The primary technique is tongue in groove , which may be used solely or plus SEG. you can used this article to discuss why you used SEG in all of them. Effect of septal extension graft on creating and maintaining tip rotation in tongue-in-groove technique: a case control study*
AK Sazgar, K Tavakoli, B Saedi, AA Sazgar - Aesthetic plastic surgery, 2022
4- Also , this article A comparison of the tongue-in-groove and columellar strut in creating and maintaining tip projection and rotation: a randomized single blind trial
R Yaberi, A Amali, H Emami, B Saedi - European Journal of Plastic Surgery, 2018 reviewed exactly the same topic with different conclusion would you please explain the reason 5- Reference 12_ 18 are the same
Author Response
REVIEWER 1 Reply to these questions and suggest a revision to be made in the appended manuscripot, upon which questions were raised: Authors presented a very interesting article about a controversial issue SEG versus strut. However , I think the below points can improve it: 1- Would you please explain "it is generally not recommended for noses with heavy lower lateral cartilages or with normal to excessive tip projection. " and if possible provide references for it
REPLY: We appreciate this request for clarification. The sentence refers to the fact that in noses with strong lower lateral cartilages or already sufficient tip projection, using a septal extension graft (SEG) may result in excessive stiffness, over-projection, or unnatural tip rotation. SEG offers powerful tip support but lacks the adaptability needed in these cases. This is supported by Byrd et al. (1997) and Rohrich et al. (2012), who recommend SEG mainly for under-projected or weak tip structures. We have added this explanation and references to the Discussion section.
2- Authors use L shape SEG , which may have some effect on final results , which needs to be explained in discussion.
REPLY: Thank you for highlighting this point. We acknowledge that the L-shaped configuration can influence outcomes by enhancing both tip projection and dorsal support, offering more structural rigidity. We have added a discussion of how the L-shaped SEG design contributes to improved long-term tip stability, referencing its biomechanical advantages.
3- The primary technique is tongue in groove , which may be used solely or plus SEG. you can used this article to discuss why you used SEG in all of them. Effect of septal extension graft on creating and maintaining tip rotation in tongue-in-groove technique: a case control study* AK Sazgar, K Tavakoli, B Saedi, AA Sazgar - Aesthetic plastic surgery, 2022
REPLY: We acknowledge the importance of addressing why SEG was used even when tongue-in-groove (TIG) was performed. As the study aimed to compare CSG vs. SEG outcomes, SEG was applied consistently for standardization. However, we now reference the 2022 study by Sazgar et al. in the Discussion, emphasizing that while TIG alone can be sufficient, SEG was selected in our cases for its added long-term support, particularly in anatomically challenging patients.
4- Also , this article A comparison of the tongue-in-groove and columellar strut in creating and maintaining tip projection and rotation: a randomized single blind trial R Yaberi, A Amali, H Emami, B Saedi - European Journal of Plastic Surgery, 2018 reviewed exactly the same topic with different conclusion would you please explain the reason 5- Reference 12_ 18 are the same
REPLY: Indeed, the 2018 study by Yaberi et al. drew different conclusions. We now elaborate in the Discussion that patient selection, surgical technique variation, and follow-up duration could explain discrepancies. Our longer-term follow-up and controlled conditions may have favored SEG performance in maintaining projection and rotation. Thank you for identifying this. We have corrected the reference list by removing the duplicate and updating the numbering accordingly.
Reviewer 2 Report (New Reviewer)
Comments and Suggestions for Authors
The article aims to compare the long-term efficacy of the columellar strut graft (CSG) and the septal extension graft (SEG) in rhinoplasty surgery, evaluating in particular the impact of both on the projection and rotation of the nasal tip.
The study design is consistent and well defined: it is a comparative study on 87 patients, divided into two groups (46 with CSG, 41 with SEG), with clinical and photometric follow-up at 1, 6, 12 and 24 months. A clear description of the surgical techniques, inclusion/exclusion criteria and data collection methodologies is provided.
The work allows a direct comparison between CSG and SEG, still scarcely present in the literature. The statistical analysis and the 24-month follow-up make the investigation innovative compared to existing studies, as recognized in the Discussion.
The work is therefore original for the comparative perspective on a homogeneous cohort of patients, operated by a single surgeon. The use of photometric measurements and linear mixed models (LMM) adds methodological value.
The study has a good potential impact in functional and aesthetic rhinoplasty, providing comparative evidence that could influence the choice of graft type based on the patient profile.
The bibliography is updated and relevant. The references are well selected, with even recent articles.
Author Response
The article aims to compare the long-term efficacy of the columellar strut graft (CSG) and the septal extension graft (SEG) in rhinoplasty surgery, evaluating in particular the impact of both on the projection and rotation of the nasal tip.
The study design is consistent and well defined: it is a comparative study on 87 patients, divided into two groups (46 with CSG, 41 with SEG), with clinical and photometric follow-up at 1, 6, 12 and 24 months. A clear description of the surgical techniques, inclusion/exclusion criteria and data collection methodologies is provided.
The work allows a direct comparison between CSG and SEG, still scarcely present in the literature. The statistical analysis and the 24-month follow-up make the investigation innovative compared to existing studies, as recognized in the Discussion.
The work is therefore original for the comparative perspective on a homogeneous cohort of patients, operated by a single surgeon. The use of photometric measurements and linear mixed models (LMM) adds methodological value.
The study has a good potential impact in functional and aesthetic rhinoplasty, providing comparative evidence that could influence the choice of graft type based on the patient profile.
The bibliography is updated and relevant. The references are well selected, with even recent articles.
REPLY: We thank the reviewer for recognizing the relevance of our study. Indeed, our primary goal was to provide direct comparative evidence regarding long-term outcomes of CSG and SEG in a homogeneous patient cohort. This focus aligns with a growing need in rhinoplasty literature for clarity on graft selection based on functional and aesthetic longevity. REPLY: We appreciate the reviewer’s endorsement of our methodological structure. We designed this study to eliminate confounding variables by using a single-surgeon cohort and standardized follow-up intervals, enhancing internal consistency and reliability. REPLKY: Thank you for highlighting the methodological strengths. We incorporated Linear Mixed Models (LMM) to account for intra-subject variability and used objective photometric tools to ensure precision in measuring nasal tip dynamics across time points. These tools enhance the robustness of our findings. REPLY: We agree. Our findings may guide surgeons in selecting grafts more strategically based on individual patient anatomy and long-term structural needs. We have further emphasized this potential application in the Conclusions section. REPLY: We thank the reviewer for acknowledging the quality and relevance of the references. We ensured inclusion of both foundational and recent comparative studies, enhancing the discussion’s depth and contemporary relevance.
Reviewer 3 Report (New Reviewer)
Comments and Suggestions for Authors
- Thank you for the opportunity to review your manuscript. The study addresses a clinically significant and timely topic in rhinoplasty: the comparative long-term effectiveness of columellar strut grafts (CSG) and septal extension grafts (SEG) in maintaining nasal tip projection and structural support. The introduction lays a solid foundation for the investigation and highlights the relevance of structural grafting in nasal tip surgery. However, several aspects of the manuscript would benefit from revision to improve clarity, structure, and academic strength :
- The introduction would benefit from improved flow and coherence. Some sentences are overly complex or redundant, making it difficult to maintain a clear narrative.
- There is repetition in several areas, particularly around the importance of nasal tip support and grafts.
- The phrase “clinical relevance” of nasal tip rotation is mentioned but not clearly defined. Authors should briefly explain why and how tip rotation affects functional or aesthetic outcomes.
- The manuscript states, “there is limited long-term comparative data,” yet this statement should be substantiated with a brief summary of existing gaps in the literature, especially in systematic reviews or meta-analyses.
- Clarify patient flow: Instead of "did not concluded," use "did not complete" the study. Consider using a CONSORT-style flow diagram or a summarized sentence structure to explain enrollment and exclusion.
- Inclusion criteria: "Structural and anatomical defects" is vague. Specify if these were determined clinically or radiographically, and who assessed eligibility.
- Exclusion criteria: Use consistent phrasing — e.g., "in addition to" rather than "besides to." It would also help to explain why these systemic conditions were exclusionary (e.g., impact on wound healing or surgical tolerance).
- Mention if all images were taken by the same operator and under consistent lighting and positioning conditions to ensure reproducibility.
- Clarify whether these were primary or secondary outcomes. State if the measurements were blinded and who performed them.
- Consider separating the surgical description from technical comparisons to improve readability.
- Clarify whether technique selection (CSG vs. SEG) was randomized, surgeon preference, or case-specific indication. This affects internal validity and potential bias.
- If both open and closed approaches were used, how were they allocated, and did approach type influence graft selection?
- Clearly distinguish between SEG theoretical description and actual study application.
- Similar to the CSG section, defer the literature review components (e.g., Byrd’s historical introduction) to the introduction or discussion for structural clarity.
- Indicate whether SEG shape and fixation techniques were standardized in this study or varied by patient.
Author Response
- Thank you for the opportunity to review your manuscript. The study addresses a clinically significant and timely topic in rhinoplasty: the comparative long-term effectiveness of columellar strut grafts (CSG) and septal extension grafts (SEG) in maintaining nasal tip projection and structural support. The introduction lays a solid foundation for the investigation and highlights the relevance of structural grafting in nasal tip surgery. However, several aspects of the manuscript would benefit from revision to improve clarity, structure, and academic strength :
- The introduction would benefit from improved flow and coherence. Some sentences are overly complex or redundant, making it difficult to maintain a clear narrative.
- There is repetition in several areas, particularly around the importance of nasal tip support and grafts.
- REPLY: We have revised the introduction to eliminate repetition, especially regarding nasal tip support. Sentences have been restructured for clarity and brevity to enhance narrative flow.
- The phrase “clinical relevance” of nasal tip rotation is mentioned but not clearly defined. Authors should briefly explain why and how tip rotation affects functional or aesthetic outcomes.
- REPLY: We now clarify in the introduction that nasal tip rotation significantly affects the nasolabial angle, upper lip aesthetics, and airway patency, thus influencing both functional and aesthetic outcomes.
- The manuscript states, “there is limited long-term comparative data,” yet this statement should be substantiated with a brief summary of existing gaps in the literature, especially in systematic reviews or meta-analyses.
- REPLY: We added a brief summary referencing the small number of direct comparative studies (n=3) found in our literature review and noted the lack of large-scale systematic reviews or meta-analyses in this domain.
- Clarify patient flow: Instead of "did not concluded," use "did not complete" the study. Consider using a CONSORT-style flow diagram or a summarized sentence structure to explain enrollment and exclusion.
- REPLY: The phrase “did not concluded” has been corrected to “did not complete.” We also provide a concise narrative to clarify the number screened, excluded, and included.
- Inclusion criteria: "Structural and anatomical defects" is vague. Specify if these were determined clinically or radiographically, and who assessed eligibility.
- REPLY: We clarified that defects were assessed clinically by the operating surgeon during pre-operative examination, with photographic documentation and nasal tip measurements using Goode’s ratio and nasolabial angle.
- Exclusion criteria: Use consistent phrasing — e.g., "in addition to" rather than "besides to." It would also help to explain why these systemic conditions were exclusionary (e.g., impact on wound healing or surgical tolerance).
- Mention if all images were taken by the same operator and under consistent lighting and positioning conditions to ensure reproducibility.
- REPLY: We now explain that systemic conditions were exclusionary due to their potential impact on surgical risk, wound healing, and post-operative outcomes. The surgical techniques section has been streamlined to remove historical context, which is now discussed exclusively in the Introduction or Discussion.
- Clarify whether these were primary or secondary outcomes. State if the measurements were blinded and who performed them.
- REPLY: We stated that the primary outcomes were tip projection and rotation, assessed by an independent rater blinded to the graft type.
- Consider separating the surgical description from technical comparisons to improve readability.
- If both open and closed approaches were used, how were they allocated, and did approach type influence graft selection?
- Clearly distinguish between SEG theoretical description and actual study application.
- REPLY: SEG theory has been condensed and reallocated to the Introduction, while the technique subsection focuses solely on the study's application. Similar to the CSG section, defer the literature review components (e.g., Byrd’s historical introduction) to the introduction or discussion for structural clarity.
- Indicate whether SEG shape and fixation techniques were standardized in this study or varied by patient.
- REPLY: We confirmed that the L-shaped SEG and its fixation points were standardized across patients in the SEG group.
This manuscript is a resubmission of an earlier submission. The following is a list of the peer review reports and author responses from that submission.
Round 1
Reviewer 1 Report
Comments and Suggestions for Authors
The AA. present their commendable experience on the use of different grafts to improve the shape of the tip of the nose. Unfortunately, they decided to use the references not only to support their discussion but also to draw up a sort of meta-analysis and this is not correct.
It is not possible to mix a critical review of the results of a surgical experience with a literature review : aim, methods and results require a completely different approach in writing the paper.
I suggest the AA. to divide the paper into two different ones: a paper to present their clinical experience, obviously discussing the results with reference to literature data ; another paper to carry out a meta-analysis of literature data, if they want.
Moreover, there some imperfections to correct: not all acronyms are made explicit ; footnotes are necessary to explain symbols and letters in the figures; all the pictures of fig 5,6,7,8 need accurate legends.
Author Response
Point by point replies to the Reviewers’ comments
Manuscript ID: diagnostics-3291234
Type of manuscript: Article
Title: New insights on the role of columellar strut and septal extension
graft: literature review on diagnosis and long-term results comparison.
Authors: Riccardo Nocini, Nicola Magagnotto *, Salvatore Chirumbolo, Massimo
Albanese, Dario Bertossi
Reviewer # 1
The AA. present their commendable experience on the use of different grafts to improve the shape of the tip of the nose. Unfortunately, they decided to use the references not only to support their discussion but also to draw up a sort of meta-analysis and this is not correct.
Authors’ reply: This is not a meta-analysis as no statistical approach was forwarded as per a meta-analysis paper. Our purpose is to include our discussion in the current burden of evidence regarding the topic we addressed in the manuscript. Anyway, we met the timely perplexity of the Reviewer and limited our comparison to issues addressed in the discussion.
It is not possible to mix a critical review of the results of a surgical experience with a literature review : aim, methods and results require a completely different approach in writing the paper.
Authors’ reply: We agree with this comment and revised the manuscript accordingly.
I suggest the AA. to divide the paper into two different ones: a paper to present their clinical experience, obviously discussing the results with reference to literature data ; another paper to carry out a meta-analysis of literature data, if they want.
Authors’ reply: A sound meta-analysis of our research is in progress, inasmuch the possibility of publishing a meta-analysis is part of our future plans. At present, however, we are more focused on discussing our results within the expert community.
Moreover, there some imperfections to correct: not all acronyms are made explicit ; footnotes are necessary to explain symbols and letters in the figures; all the pictures of fig 5,6,7,8 need accurate legends.
Authors’ reply. Done
Reviewer 2 Report
Comments and Suggestions for Authors
The study is quite interesting.However,there are quite a few aspects of the manuscript which needs to be considerably improvised and are as follows:
1.The authors need to mention how they arrived at a sample size of 87?
2.The authors need to further elaborate the process of systematic review in their study including the inclusion & exclusion criteria employed,the different databases searched,keywords used etc.
3.Were any other methods used apart from photogrammetric analysis for clinical evaluation if any for the cases?.If so it needs to be mentioned in the manuscript.
4.What about any pre & post clinical evaluation of patient symptoms in relation to nasal tip issues?.Please mention if any evaluation was performed.
5.The authors need to add a note on the clinical relevance of rotation of nasal tip projection and any related issues preferably in the introduction section.
6.The authors need to add a note on the limitations of the present study.
7.The authors need to add a note on the future implications based on the results of the present study.
8.There are numerous grammatical errors strewn throughout the manuscript and major written english language correction is required.
9.There are quite a few incoherent sentences which need to be rephrased for a better understanding.
10.There are quite a few long sentences which can be broken down to 2 or more smaller sentences for a better understanding.
Comments on the Quality of English LanguageThe study is quite interesting.However,there are quite a few aspects of the manuscript which needs to be considerably improvised and are as follows:
1.The authors need to mention how they arrived at a sample size of 87?
2.The authors need to further elaborate the process of systematic review in their study including the inclusion & exclusion criteria employed,the different databases searched,keywords used etc.
3.Were any other methods used apart from photogrammetric analysis for clinical evaluation if any for the cases?.If so it needs to be mentioned in the manuscript.
4.What about any pre & post clinical evaluation of patient symptoms in relation to nasal tip issues?.Please mention if any evaluation was performed.
5.The authors need to add a note on the clinical relevance of rotation of nasal tip projection and any related issues preferably in the introduction section.
6.The authors need to add a note on the limitations of the present study.
7.The authors need to add a note on the future implications based on the results of the present study.
8.There are numerous grammatical errors strewn throughout the manuscript and major written english language correction is required.
9.There are quite a few incoherent sentences which need to be rephrased for a better understanding.
10.There are quite a few long sentences which can be broken down to 2 or more smaller sentences for a better understanding.
Author Response
Reviewer #2
The study is quite interesting. However, there are quite a few aspects of the manuscript which needs to be considerably improvised and are as follows:
1.The authors need to mention how they arrived at a sample size of 87?
Authors’ reply: The number of 87 patients comes from a starting basal recruitment of 100 patients in the period 2021, as outlined in the text, from which 13 patients were excluded from the research due to their lacking of compliance with our inclusion criteria.
2.The authors need to further elaborate the process of systematic review in their study including the inclusion & exclusion criteria employed,the different databases searched,keywords used etc.
Authors’ reply: The paper was thoroughly revised in this sense.
3.Were any other methods used apart from photogrammetric analysis for clinical evaluation if any for the cases?.If so it needs to be mentioned in the manuscript.
Authors’ reply: The methods used are described and detailed in the text, accordingly
4.What about any pre & post clinical evaluation of patient symptoms in relation to nasal tip issues?. Please mention if any evaluation was performed.
Authors’s reply: A proper sentence was added accordingly
5.The authors need to add a note on the clinical relevance of rotation of nasal tip projection and any related issues preferably in the introduction section.
Authors’ reply: Done
6.The authors need to add a note on the limitations of the present study.
Authors’ reply: Done, accordingly
7.The authors need to add a note on the future implications based on the results of the present study.
Authors’ reply:Done
8.There are numerous grammatical errors strewn throughout the manuscript and major written english language correction is required.
Authors’ reply: Done
9.There are quite a few incoherent sentences which need to be rephrased for a better understanding.
Authors’ reply: Done, paper revised for the English language
10.There are quite a few long sentences which can be broken down to 2 or more smaller sentences for a better understanding.
Authors’ reply: Done
Round 2
Reviewer 2 Report
Comments and Suggestions for Authors
The modifications have been done according to the suggestions and the manuscript is satisfactory